# Aeroponic cultivation of lettuce: Unravelling varietal performance and trait interrelationships for enhanced productivity

Anand Sahil[1,2], S. R. Singh[2], Lalu Prasad Yadav [ORCID][1]*, A. K. Singh[1], Vikas Yadav[1], Sanjay Kumar[3], N. K. Meena[4], Prashant Kaushik[3], Prakash Mahala[5]

**1** ICAR- Central Horticultural Experiment Station (CIAH RS), Godhra, Gujarat, India, **2** ICAR-Central Institute for Subtropical Horticulture, Rehmankhera, Lucknow, Uttar Pradesh, **3** Chaudhary Charan Singh Haryana Agricultural University, Hisar, Haryana, India, **4** College of Horticulture and Forestry, Agriculture University, Kota, Jhalawar, Rajasthan, India, **5** Punjab Agricultural University, Regional Research Station, Abohar, India

* yadavlaluprasad682@gmail.com

## Abstract

Aeroponic systems offer a sustainable and efficient platform for cultivating high-quality leafy greens, such as lettuce (*Lactuca sativa* L.). This study investigated the performance of five distinct lettuce cultivars ('Summer Star', 'Grand Rapid', 'Tango', 'Bingo', and 'Black Rose') within a controlled aeroponic environment to identify superior varieties and elucidate the intricate relationships among key agronomic traits. A comprehensive suite of statistical analyses was employed, including Analysis of Variance (ANOVA), Pearson correlation analysis, Principal Component Analysis (PCA), and genetic variability assessment using GCV and PCV. Results revealed significant differences among cultivars for growth parameters, yield components, and quality attributes. Notably, 'Bingo' exhibited the highest total leaf weight, while 'Summer Star' demonstrated superior yield per hectare. Correlation analysis highlighted strong positive associations between plant height, leaf area index, and chlorophyll content, whereas yield exhibited a negative correlation with total chlorophyll content. PCA identified key underlying factors contributing to the observed variation, with the first two principal components explaining 86.13% of the total variance, underscoring the importance of leaf morphology and chlorophyll concentration in driving aeroponic growth and productivity. The findings underscore the potential of aeroponics to contribute to global food security by enhancing the productivity and quality of leafy greens.

## 1. Introduction

Lettuce (*Lactuca sativa* L.), a member of the Asteraceae family originating from the Mediterranean region, It is globally recognized as a leading leafy salad vegetable, surpassing all other salad crops in cultivation area and production [1]. Green leafy

**Data availability statement:** All relevant data are provided within the article.

**Funding:** The author(s) received no specific funding for this work.

**Competing interests:** The authors declare that they have no financial or non-financial competing interests relevant to this study.

vegetables are highly valued in diets worldwide for their versatility in culinary uses and exceptional nutritional benefits [2]. It is being popular in winter season in subtropical areas of India and fetching a good price to the growers in metropolitan cities [3].

As lifestyle-related diseases including cancer, diabetes, obesity, and cardiovascular ailments become more common, there is an increasing need for nutrient-dense dietary options in both developed and developing countries. Furthermore, in developing and underdeveloped nations with limited access to nutrient-rich meals, particularly for those with low purchasing power, child mortality and malnutrition continue to be major problems [4]. Essential vitamins A, K, C, and numerous B vitamins are abundant in lettuce, and its abundance of secondary metabolites, such as flavanols, carotenoids, phenolic compounds, and folate, has been associated with a number of health advantages [5,6]. Due to its low calorie, fat, and sodium content, this popular salad crop is highly valued. Red leaf lettuce stands out among the other varieties of lettuce due to its high anthocyanin content, which adds to its antioxidant qualities [7].

Compounds called polyphenols have been shown to have antioxidant properties and to scavenge free radicals, which may help prevent cardiovascular and cancerous conditions [8,9]. According to Adesso et al. [10], extracts from lettuce leaves can lessen oxidative stress and inflammation in murine monocyte-macrophage cells by lowering the production of nitric oxide and reactive oxygen species. According to Kim et al. [11], the leaves are used in a variety of foods, including salads, wraps, sandwiches, soups, and processed meals.

Soilless cultivation methods, encompassing aeroponics, aquaponics, and hydroponics, stand out as innovative agricultural approaches geared towards achieving greater productivity with fewer resources. Hydroponic systems' automation and control have reduced human contact in addition to sensing and monitoring [12]. Given that there will likely be nine billion people on the planet by 2050, food security is a major issue of the new millennium and maybe the biggest obstacle facing the agriculture industry [13]. Global output will need to rise by almost 70% from 2007 levels to meet the demand for food. 75% of the world's population is expected to live in urban regions due to the world's increasing urbanization [14]. By 2100, there will be 11 billion people on the planet, and this is essential to supplying their food needs [15].

A developing technology called aeroponics offers a viable substitute for conventional cultivation techniques by growing plants in an enclosed chamber with their root systems exposed to a nutrient mist [16]. This method has become popular for producing several vegetable crops, particularly those where the roots are the main part that can be harvested, such as lettuce, cucumber, melon, tomato, herbs, and potatoes. Aeroponics has also been used in space missions, where it ensures the best possible control of plant growth factors, making it an essential life support system.

By maximizing root aeration, aeroponics has been demonstrated to dramatically increase plant growth rates. Plants that are suspended in the air have access to all the atmospheric oxygen, which speeds up development for both the stem and root systems [17]. The oxygen-rich environment that aeroponics provides is largely responsible for the success of plant development in this method. Better growth

management is made possible by the incorporation of gas supply systems, which further improves the ability to regulate the root-zone atmosphere [18].

The size of the droplets and the frequency of root exposure to the nutrient solution are two variables that can affect the amount of oxygen available in aeroponic systems. While fine droplets may encourage excessive root hair growth but prevent the establishment of lateral roots, which are necessary for sustained plant growth, large droplets may decrease oxygen availability [19]. Furthermore, aeroponic systems may produce large amounts of tiny tubers in a single generation, eliminating the requirement for field production and allowing for sequential harvesting. Significant time and money savings are possible with this method [20].

Aeroponic systems provide many advantages over conventional cultivation. These include the lack of soil-borne pathogens, easier-to-manage pest control, effective fertilizer use, lower labor costs, fewer risks from environmental factors, higher yields and better quality, accurate nutrient control, and increased resilience to problems like tilling, weeding, and spray watering. In addition to reducing fertilizer use and enabling the cultivation of some plants out of season, aeroponics also enhances yields early in colder seasons and keeps nutrients from leaking into the environment [21].

Additionally, hydroponic systems—including aeroponics—provide a workable answer to a number of environmental problems. According to Avgoustaki and Xydis [22], hydroponic farming methods can help address problems such increasing $CO_2$ levels, water pollution, overuse of fertilizer, and the loss of arable land. Comparing the performance of various lettuce kinds cultivated in an aeroponic system, evaluating their growth, production, and quality, is the primary goal of this study.

## 2.0 Materials and methods

### 2.1 Description of experimental site and environmental conditions

In order to identify lettuce varieties that yield higher quality produce in aeroponic systems within subtropical climates, the experiment was laid out in a Randomized Block Design in two factor with three replications at ICAR-Central Institute for Subtropical Horticulture, Rehmankhera Lucknow experimental farm, throughout the period of 2021–2022. The experimental site is located between 26° 45' to 27° 10' N latitude and 80° 30' to 80° 5' E longitude, with an elevation of 123 meters above sea level. The experimental site's subtropical climate features hot, dry summers and cool, dry winters, with the southwest monsoon typically arriving from July to September. The average maximum temperature in May and June ranges between 40°C to 45°C, while under polyhouse conditions, the maximum and minimum temperatures during November to February range from 23°C to 29°C and 7°C to 13°C, respectively. Additionally, the maximum relative humidity during the cropping season is between 82% to 92%, with the minimum relative humidity varying from 45% to 68%, as depicted in Fig 1.

### 2.1.1. Statement on experimental research and field studies on plants

The either cultivated or wild-growing plants sampled comply with relevant institutional, national, and international guidelines and domestic legislation of India.

### 2.2 Selection of varieties, growing media, and sowing strategies

In the first week of October, five lettuce varieties such as Summer star, Grand rapid, Tango, Bingo, and Black rose. The seeds were procured from ICAR- Central Institute of Temperate Horticulture, Srinagar, India. The seeds were sown in seedling trays with 98 cavity holes, each filled with a prepared growing medium consisting of cocopeat, vermiculite, and perlite in a 3:1:1 (v/v) ratio. The mixture was sterilized with a 0.02% hydrogen peroxide solution. After 35days, when the plants attained 4–5 leaves, they were transplanted into an aeroponic system within a polyhouse (Fig 2).

 

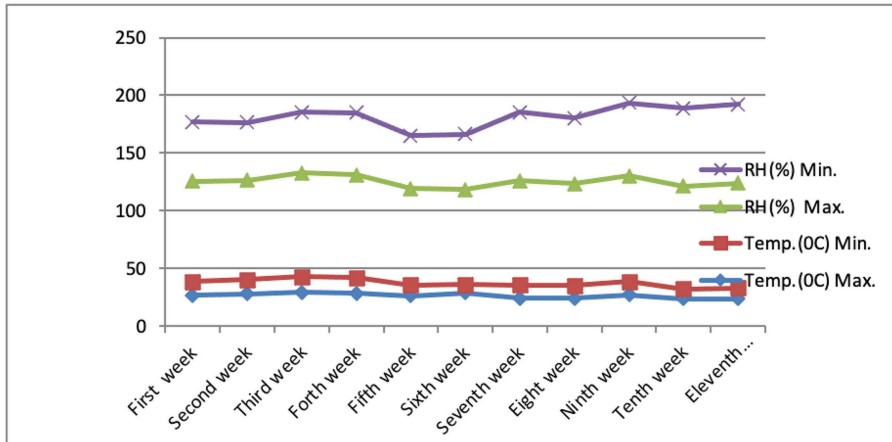

**Fig 1. Mean weekly weather data from November 2021 to February 2022.**

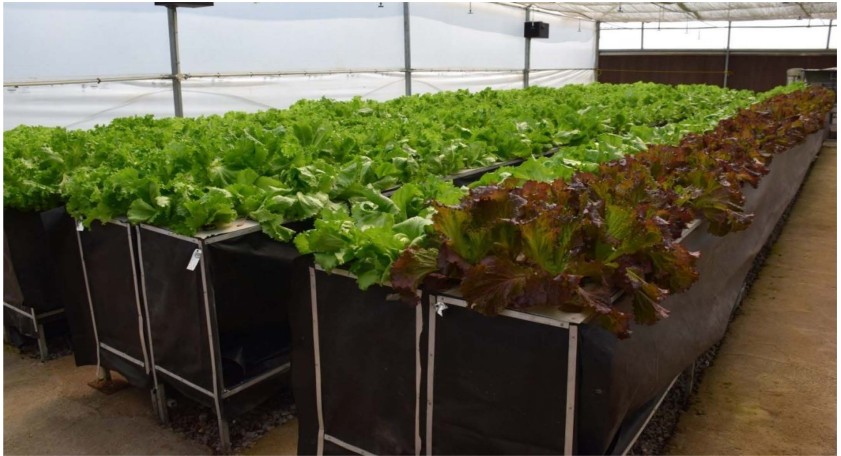

**Fig 2. Experimental site and view of aeroponics system of lettuce.**

## 2.3 Lettuce growing system using aeroponic technology

The aeroponics unit was installed in a polyhouse with automated atmospheric control. The roof is covered with a 200-micron UV-stabilized polyethylene sheet, while the sidewalls are made of either a 200-micron UV-stabilized polyethylene sheet or insect-proof netting. Inside, silver shade nets, fans, a pad, and a misting system were used to maintain optimal conditions for the experiments. The units were constructed in an A-frame design, measuring 1.4 meters in width, 1.4 meters in height, and 6 meters in length. The planting density was standardized at 25 plants m², with a spacing about 20 cm × 20 cm between each plant. After 30 days, the plants were given nutrient solutions CISH A and CISH B (https://cish.icar.gov.in/leafy_vegetables_solution.php) having 3x strength one litre each, under aeroponics system. One litter of CISH A and CISH B nutrient solutions was added to each water tank in the aeroponic system. The misting session, rich in nutrients, automatically ran for one minute every 30 minutes. In this auto-aeroponic system, the nutrient solution temperature was maintained between 15−18°C, with electrical conductivity between 1.0–1.2 mS/cm and a pH range of 5.5 to 6.5 throughout the growing season.

## 2.4 Yield and yield attributing traits

At the time of harvesting, ten fully grown plants from each treatment were randomly selected in each replication to record the data. Vegetative parameters, including plant height, were recorded following a standard procedure. The height of fully developed plants was measured from ground level to the leaf apex in cm, and the average was calculated. Plant weight was recorded using an electronic weighing machine, and the average weight was calculated in grams. Root length was measured in centimetres with a measuring scale, and the average value was computed. The plant stem diameter was measured just above ground level using a Mitutoyo Digital Vernier Calliper, and the average value was recorded in mm. The total number of leaves was counted on fully grown plants, and the average was calculated per plant. Plant shoot weight was observed by removing the roots, weighing the shoots with a weighing machine, and calculating the average weight in grams. Leaf width was measured in cm at the middle portion of the leaf using a measuring scale, and the average value was calculated. Leaf length was measured from the base to the top of the leaf in cm, and the average value was determined. Leaf weight was assessed by collecting the leaves from the plants, weighing them with a scale, and calculating the average weight in grams [3] and at harvest, the average weight of 10 plants was measured, and the yield was converted to quintals per hectare.

## 2.5 Quality parameters

For dry matter content of the leaves was determined using the oven-dry method. A 25 g fresh leaf sample was dried in an oven at 60°C until it reached a constant weight, and it was expressed as percentage [23]. Ascorbic acid content was determined using the 2,6-dichlorophenol-indophenol titration method, following the formula recommended by AOAC [24]; Dolma et al [25,4,26,27]. The total carotenoid content of leaves was determined by crushing a 5g sample through mortar and pestle adding 15 ml acetone and. The mixture was filtered, and then separate carotenoids in a separation funnel with 10% NaCl solution and make up volume was adjusted with acetone in petroleum ether, and color intensity was measured at 452 nm [28,26]. Chlorophyll "a", Chlorophyll "b" and Total Chlorophyll was determined at wavelengths of 600nm, 642.5nm, and 660nm and chlorophyll 'a' was calculated using the formula: Chlorophyll 'a' = (9.93 x OD at 660nm) – (0.777 x OD at 642.5nm). Chlorophyll 'b' was determined using the formula= (17.6 x OD at 642.5nm) – (2.81 x OD at 660nm). Total chlorophyll was determined using the formula= (7.12 x OD at 660 nm) + (16.8 x OD at 642.5nm) [29,30].

## 2.6 Statistical analysis

A thorough analysis of variance (ANOVA), as described by Fisher [31], was carried out to rigorously assess the performance of five different lettuce cultivars grown under aeroponic conditions. This approach enabled us to identify notable variations in the cultivars' growth, yield, and quality features. The interrelationships between the assessed attributes were clarified using [32], performed after ANOVA. This analysis revealed possible antagonistic or synergistic interactions and gave insights into the trait connections. Principal Component Analysis (PCA), a technique refined by Hotelling [33], was then employed to identify important underlying elements contributing to the observed variation and to simplify the multidimensional data into principal components that represented the most important patterns.

Furthermore, the Genetic Coefficient of Variation (GCV) and Phenotypic Coefficient of Variation (PCV), as discussed by Lush [34], were used to quantify genetic variability. These coefficients provided a comparative view of how genetic and environmental variables affect the expression of traits. Each variety's performance was also assessed directly using yield per hectare, total leaf weight, and leaf count, offering a useful evaluation of their agronomic potential. In order to evaluate the symmetry and tailedness of each trait's distributional characteristics, skewness and kurtosis were computed, following the methods outlined by Bulmer [35].

All statistical analyses were performed using the R statistical computing environment (version 4.3.1; R Core [36]) with specialized packages, including lme4 [37] for linear mixed-effects models used in ANOVA, corrplot [38] for correlation matrix visualization, and factoextra [39] for PCA.

 

## 3. Results

### 3.1 Correlation among the traits *Lactuca sativa* L. accessions grown in an aeroponic system

Significant variations exist between the *Lactuca sativa* L. accessions grown in an aeroponic system for the majority of the growth and quality attributes, according to the findings of the ANOVA (Table 1). The very significant differences (p < 0.001***) in parameters including plant height (PH), plant weight (PW), total leaf weight (TLW), specific leaf area (SD), and yield (Y) reveal that these variables are highly responsive to environmental influences and genetic variation. As evidence of their potential for selection and improvement, other characteristics such as root length (RL), leaf area index (LAI), and chlorophyll content (Chyll a, Chyll b, Total Chyll) also show considerable variation (p < 0.001). On the other hand, characteristics such as leaf length (LL) did not differ significantly (p = 0.4146, NS), suggesting that under the aeroponic system, there was less variation in this attribute across the accessions. The intermediate significance (p < 0.05 or p < 0.01) of qualities like leaf width (LW), number of leaves per plant (NL/P), and leaf water content (ASW) indicates that they also contribute to variability, but less so than the highly significant traits.

The correlation coefficients for various traits of *Lactuca sativa* L. during the 2021–2022 period was determined using half-triangle correlation matrices, as summarized in Fig 3. In this study, plant height (PH) exhibited a very strong positive correlation with chlorophyll a (0.90) and LAI (0.93). Additionally, PH showed positive correlations with chlorophyll b (0.77), RL (0.72), total chlorophyll (0.66), ASW (0.48), NL/P (0.57), LL (0.55), and TC (0.39). Conversely, negative correlations were observed between PH and variables such as VC (0.16), Y (0.39), ARW (0.68), SD (0.25), TLW (0.45), LW (0.73), and PW (0.42). PW showed a very strong and significant positive correlation with TLW (1.00), Y (0.94), and SD (0.88). Additionally, PW exhibited positive correlations with ARW (0.72), LW (0.64), and ASW (0.43). Conversely, PW demonstrated negative correlations with total chyll (−0.39), Chyll b (−0.50), Chyll a (−0.30), TC (−0.63), VC (−0.52), LAI (−0.40), RL (−0.06), NL/P (0.11), and LL 0.13. A strong positive correlation was observed of leaf length (LL) in respect of NL/P (0.81), followed by RL (0.68), LAI (0.61), ASW (0.60), chyll a (0.51), chlyll b (0.33), and SD (0.29). However, LL showed negative correlations with VC (−0.56), TC (−0.01), Y (0.10), ARW (−0.07), LW (0.02), TLW (0.15), and total chyll (0.15).

**Table 1. ANOVA for growth and quality attributes in *Lectuca sativa* L. accessions grown under aeroponics system.**

| Traits | F-statistic | P-value |
|---|---|---|
| PH | 42.15 | 0.0005*** |
| PW | 189.99 | 1.2E-05*** |
| LL | 1.19 | 0.4146^NS |
| LW | 5.41 | 0.0463* |
| NL/P | 8.16 | 0.0204* |
| TLW | 157.79 | 1.91E-05*** |
| SD | 161.77 | 1.79E-05*** |
| ASW | 279.44 | 4.62E-06*** |
| ARW | 222.57 | 8.13E-06*** |
| RL | 139.83 | 2.57E-05*** |
| LAI | 42.79 | 0.0005*** |
| Y | 5142.96 | 3.22E-09*** |
| VC | 2766.05 | 1.52E-08*** |
| TC | 1835.97 | 4.22E-08*** |
| Chyll a | 187.89 | 1.24E-05*** |
| Chyll b | 616.63 | 6.43E-07*** |
| Total Chyll | 78.69 | 0.0001*** |

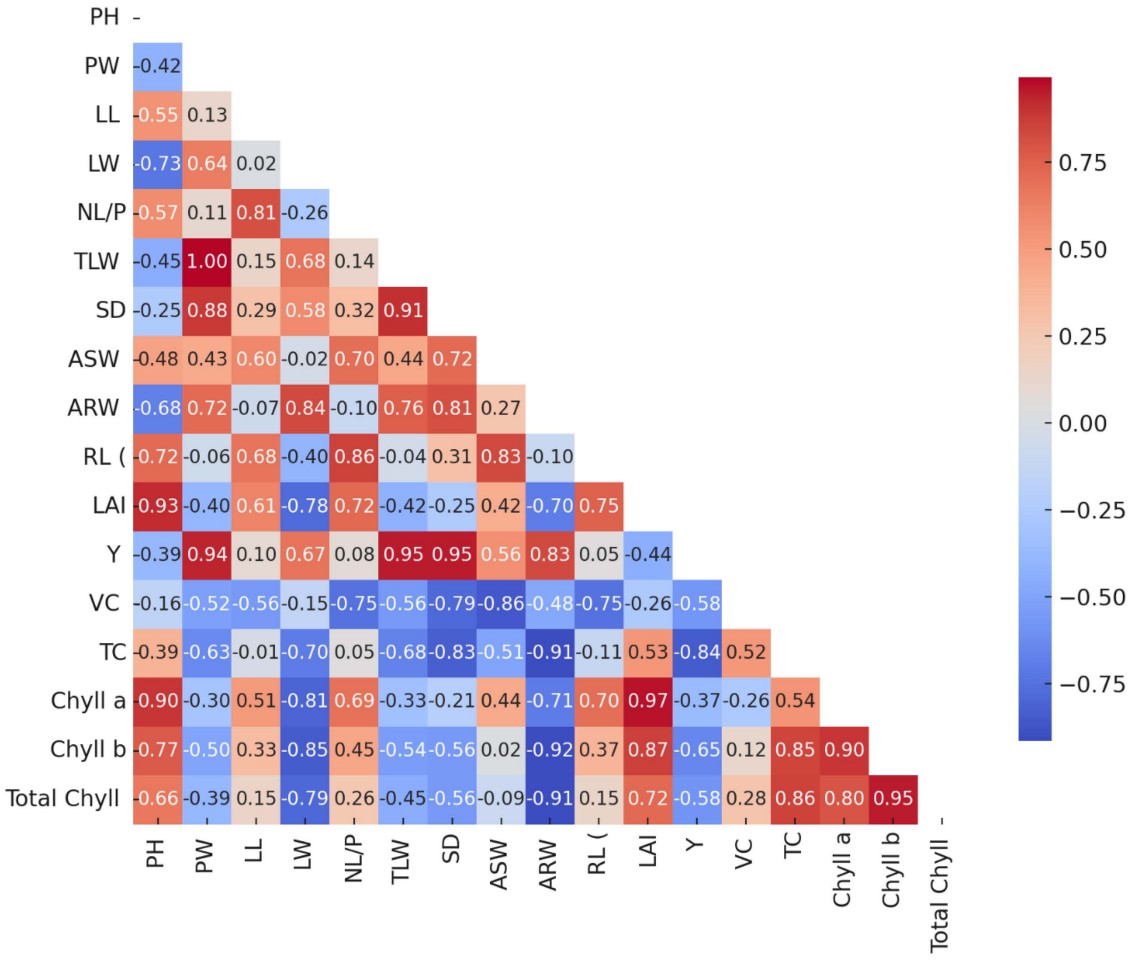

**Fig 3. Half-triangle correlation matrices among different traits of *Lectuca sativa* L. during the year 2021-2022.**

LW showed a significant positive correlation with ARW (0.84). Additionally, positive correlations were observed with TLW (0.68), Y (0.67), and SD (0.58). Conversely, LW demonstrated negative correlations with total chyll (−0.79), chyll b (−0.85), chyll a (−0.81), TC (−0.70), VC (−0.15), LAI (−0.78), RL (−0.40), ASW (−0.02), and NL/P (−0.26). A strong positive correlation of NL/P was observed with RL (0.86), LAI (0.72), and ASW (0.70). Additionally, positive correlations were noted with chlorophyll a (0.69), chlorophyll b (0.45), and SD (0.32). In contrast, NL/P exhibited negative correlations with VC (−0.75), ARW (−0.10), Y (0.08), TC (0.05), total chyll (0.26), and TLW (0.14). TLW showed a strong positive correlation with yield (Y) (0.95) and SD (0.91). Additionally, TLW exhibited positive correlations with ARW (0.76) and ASW (0.44). However, negative correlations were observed with total chyll (−0.68), VC (−0.56), chyll a (−0.33), chyll b (−0.54), total chyll (−0.45), LAI (−0.42), and RL (−0.04). SD demonstrated a strong positive correlation with Y (0.95) and also showed positive correlations with ARW (0.81), ASW (0.72), and RL (0.31). On the other hand, SD exhibited negative correlations with total chyll (−0.56), chyll b (−0.56), chyll a (−0.21), TC (−0.83), VC (−0.79), and LAI (−0.25). ASW showed a significant positive correlation with RL (0.83) and also exhibited positive correlations with chyll a (0.44), Y (0.56), LAI (0.42), and ARW (0.27). Conversely, ASW demonstrated negative correlations with VC (−0.86), total chyll (−0.51), chyll b (0.02), and total chyll (−0.09). ARW recorded a strong positive correlation with Y (0.83). However, ARW was negatively correlated with total chyll (−0.91), chhyll b (−0.92), chyll a (−0.71), TC (−0.91), VC (−0.48), LAI (−0.70), and RL (−0.10). RL exhibited positive

correlations with LAI (0.75), chyll a (0.70), and chyll b (0.37). However, it showed negative correlations with VC (−0.75), TC (−0.11), Y (0.05), and total chyll (0.15). LAI demonstrated a very strong positive correlation with chyll a (0.97) and chyll b (0.87). Additionally, LAI showed positive correlations with total chyll (0.72) and TC (0.53). On the other hand, it exhibited negative correlations with VC (−0.26) and Y (−0.44). A negative correlation of Y was recorded with Total chyll (−0.58), chyll b (−0.65), chyll a (- 0.37), TC (−0.84) and VC (−0.58). VC showed a significant positive correlation with TC (0.52) and a moderate positive correlation with total chyll (0.28). However, it exhibited a negative correlation with chlorophyll a (−0.26) and chlorophyll b (0.12). TC demonstrated a strong positive correlation with total chlorophyll (0.86) and chlorophyll b (0.85), followed by a moderate positive correlation with chlorophyll a (0.54). Chyll a showed a significant positive correlation with chyll b (0.90) and total chlorophyll (0.80). Similarly, chyll b exhibited a strong positive correlation with total chyll (0.95).

### 3.2 Principal component analysis

The principal component analysis (PCA) results show that the first five principal components contain the majority of the dataset's variance (Table 2 and Fig 4). In particular, PC1 and PC2 together explain 86.13% of the variance, with PC1

**Table 2. Principal principal component analysis, EV and V.**

| PC | Explained Variance (%) | Cumulative Variance (%) |
|---|---|---|
| PC1 | 51.82 | 51.82 |
| PC2 | 34.31 | 86.13 |
| PC3 | 6.64 | 92.77 |
| PC4 | 4.16 | 96.93 |
| PC5 | 2.39 | 99.32 |
| PC6 | 0.29 | 99.61 |
| PC7 | 0.24 | 99.84 |
| PC8 | 0.09 | 99.94 |
| PC9 | 0.06 | 100 |
| PC10 | 0 | 100 |

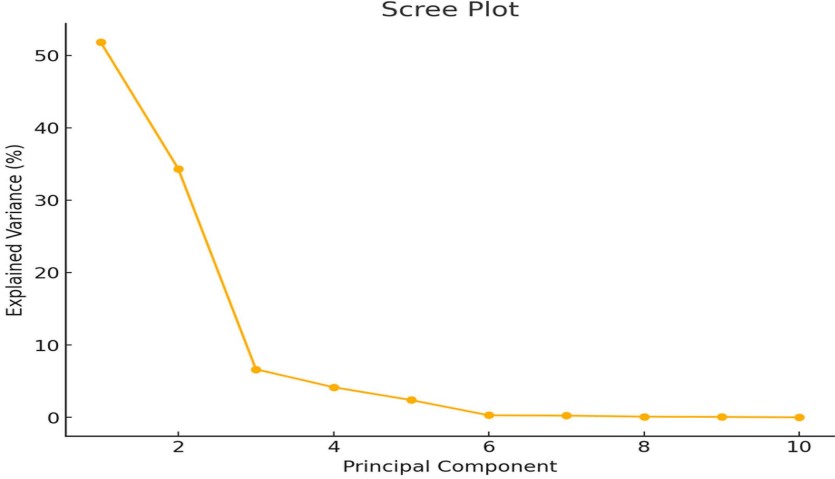

**Fig 4. PCA analysis of different traits of *Lactuca sativa* L.**

accounting for 51.82% and PC2 for 34.31%. This implies that these two elements account for the majority of the data in the dataset. The sharp decline in variance explained by later components (PC3 through PC10) emphasizes that components with higher numbers are less relevant and primarily represent noise or small fluctuations in the data.

The Table 3 presents the vector loadings of various variables (such as plant height (PH), plant weight (PW), leaf length (LL), leaf width (LW), and others) across the first five principal components (PC1 to PC5). The table also highlights the percentage of variance explained by each principal component (PC) in the analysis. The vector loadings show how much each variable contributes to each principal component. According to the PCA results, 86.13% of the variation in lettuce growth features under aeroponics can be explained by the first two main components (PC1 and PC2). Plant height, leaf breadth, and chlorophyll content all affect PC1, which accounts for 51.82% of the variance, whereas PC2 (34.31%) is related to leaf length, leaf area index, and total leaf weight. Smaller amounts of the variance are explained by subsequent components (PC3 to PC5), which concentrate on secondary features like plant weight and chlorophyll content.

The dendrogram illustrates the hierarchical clustering of five lettuce varieties: Bingo, Summer Star, Black Rose, Grand Rapid, and Tango (Fig 5). These types are categorized according to how similar they are morphologically or genetically. There are two clear clusters: the first contains Bingo and Summer Star, which show a high degree of resemblance; the second has Black Rose, Grand Rapid, and Tango, with Tango constituting a separate subgroup because of its reduced similarity to the other members of the same cluster.

**Table 3. Vector loadings and percentage explained variation.**

| PC | PH | PW | LL | LW | NL/P | TLW | SD | ASW | ARW | RL | LAI | Y | VC | TC | Chyll a | Chyll b | Total Chyll |
|---|---|---|---|---|---|---|---|---|---|---|---|---|---|---|---|---|---|
| PC1 | 0.257 | −0.256 | 0.063 | −0.301 | 0.094 | −0.267 | −0.254 | −0.046 | −0.326 | −0.102 | 0.274 | −0.284 | 0.102 | 0.292 | 0.268 | 0.316 | 0.292 |
| PC2 | −0.217 | −0.163 | −0.318 | 0.036 | −0.364 | −0.169 | −0.264 | −0.391 | −0.063 | −0.364 | −0.236 | −0.181 | 0.369 | 0.127 | −0.236 | −0.082 | −0.022 |
| PC3 | −0.102 | 0.483 | −0.034 | 0.012 | −0.017 | 0.422 | 0.101 | −0.093 | −0.147 | −0.323 | −0.016 | 0.223 | 0.106 | 0.281 | 0.141 | 0.256 | 0.455 |
| PC4 | −0.123 | −0.033 | 0.682 | 0.471 | 0.295 | 0.005 | −0.116 | −0.274 | 0.017 | −0.113 | 0.016 | −0.182 | 0.027 | 0.212 | −0.137 | 0.036 | −0.080 |
| PC5 | 0.521 | 0.019 | 0.312 | 0.229 | −0.384 | −0.035 | −0.001 | 0.238 | −0.177 | −0.113 | 0.032 | 0.186 | 0.453 | −0.277 | −0.025 | −0.076 | 0.068 |

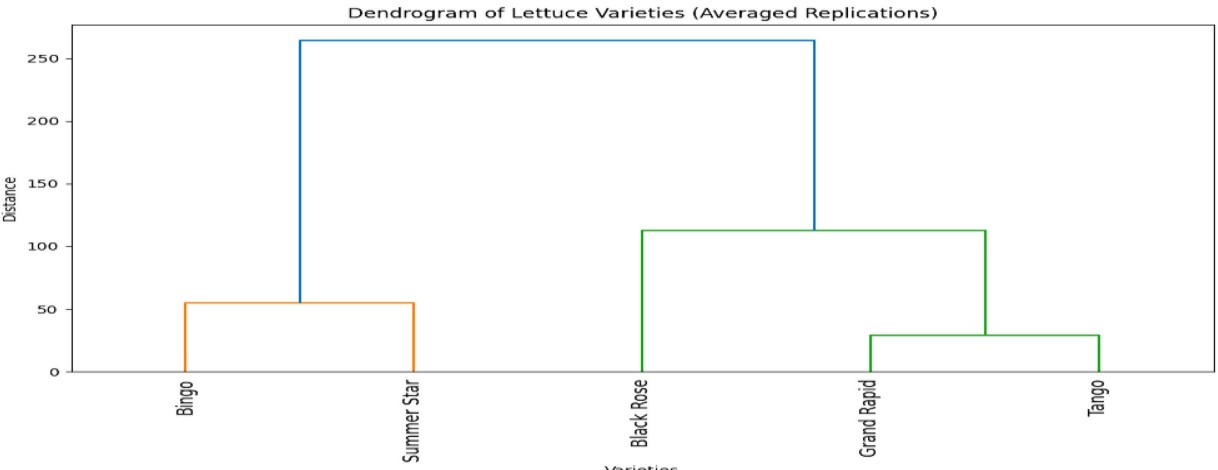

**Fig 5. Cluster analysis of *Lactuca sativa* accessions on the basis different varieties.**

## 3.3 Genetic traits

The genetic coefficient of variation (GCV), phenotypic coefficient of variation (PCV), range, and grand mean for several growth and quality traits in *Lactuca sativa* L. accessions are shown in Table 4. The traits with the highest GCV and PCV are total chlorophyll (Total Chyll), which has 41.12% GCV and 50.36% PCV, and chlorophyll a (Chyll a), which has 81.85% GCV and 100.24% PCV. Characteristics including yield (Y), plant weight (PW), and total leaf weight (TLW) have intermediate GCV and PCV values between 25.72% and 31.75%. Less variability is reflected in the decreased GCV and PCV of features such as root length (RL), leaf length (LL), leaf width (LW), and chlorophyll b (Chyll b). The grand means for the traits range from 0.53 for chlorophyll a to 259.22 for plant weight, providing a comprehensive overview of the variability in the lettuce accessions studied.

## 3.4 Comparative performance key of yield attributing traits

The performance of number of leaves, total leaf weight, and yield/ha of different varieties of lettuce under aeroponic conditions is presented in Fig 6. The number of leaves was significantly higher in the Tango variety, followed by Grand Rapid, Bingo, Summer Star, and Black Rose. The highest total leaf weight was observed in the Bingo variety, followed by Summer Star, Grand Rapid, Tango, and Black Rose. The Summer Star variety exhibited a significantly higher yield (q/ha), comparable to the Bingo variety, and was followed by Grand Rapid, Tango, and Black Rose.

## 3.5 Skewness and Kurtosis analysis

The skewness of different lettuce features is displayed in the blue bar chart (Fig 7). The data distribution is said to be right-skewed if the skewness value is positive and left-skewed if it is negative. A concentration of data points toward the lower end of the range is indicated by significant positive skewness, which is shown for variables like PW (Plant Weight) and LL (Leaf Length). Traits that indicate data points skewed towards greater values, such as VC (Vitamin C concentration) and ARW (Average Root Weight), exhibit negative skewness.

Table 4. The maximum, minimum, grand mean, GCV and PCV for growth and quality attributes in *Lectuca sativa* L. accessions.

| Character | Range (Max) | Range (Min) | Grand Mean | GCV (%) | PCV (%) |
|---|---|---|---|---|---|
| PH | 30.58 | 22.48 | 25.56 | 10.08 | 12.34 |
| PW | 355.05 | 168.35 | 259.22 | 25.75 | 31.53 |
| LL | 24.65 | 19.32 | 21.38 | 8.36 | 10.24 |
| LW | 21.05 | 15.32 | 18.52 | 10.71 | 13.12 |
| NL/P | 27.86 | 20.65 | 23.8 | 9.75 | 11.94 |
| TLW | 303.58 | 138.68 | 221.74 | 25.93 | 31.75 |
| SD | 14.52 | 11.18 | 13.32 | 8.73 | 10.69 |
| ASW | 23.58 | 10.28 | 18.17 | 25.39 | 31.09 |
| ARW | 20.58 | 11.65 | 16.67 | 24.93 | 30.53 |
| RL | 43.06 | 27.65 | 34.69 | 15.18 | 18.59 |
| LAI | 1.15 | 0.76 | 0.87 | 14.98 | 18.35 |
| Y | 213.82 | 102.64 | 164.17 | 25.72 | 31.49 |
| VC | 15.82 | 10.62 | 12.12 | 16.55 | 20.26 |
| TC | 7.98 | 4.07 | 6.11 | 23.57 | 28.87 |
| Chyll a | 1.37 | 0.19 | 0.53 | 81.85 | 100.24 |
| Chyll b | 0.19 | 0.04 | 0.1 | 51.99 | 63.67 |
| Total Chyll | 0.69 | 0.21 | 0.43 | 41.12 | 50.36 |

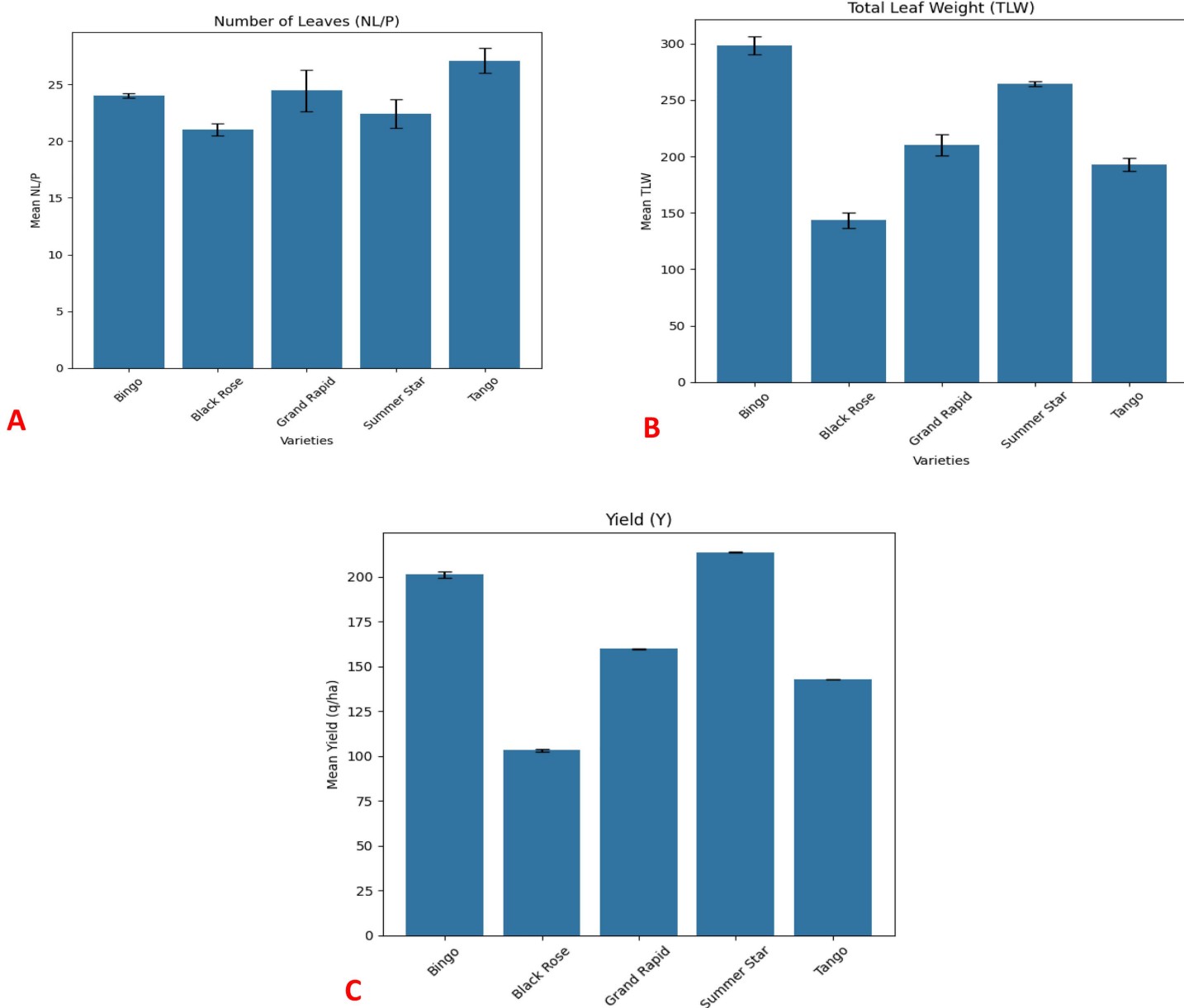

**Fig 6. Yield attributing characters of five varieties of lettuce grown under aeroponic system (A) no. of leaves, (B) total leaf weight and (C) Yield.**

Kurtosis values, which indicate how peaked or flat the distributions are, are displayed in the green bar chart. The majority of features have negative kurtosis, indicating platykurtic (flat-topped) distributions. A higher peaked distribution is indicated by traits with somewhat positive kurtosis, such as RL (Root Length). Characteristics like Total Chlorophyll, Chl a (Chlorophyll a), and ASW (Average Shoot Weight) have significant negative kurtosis, suggesting a broader data spread with fewer extreme values. K.

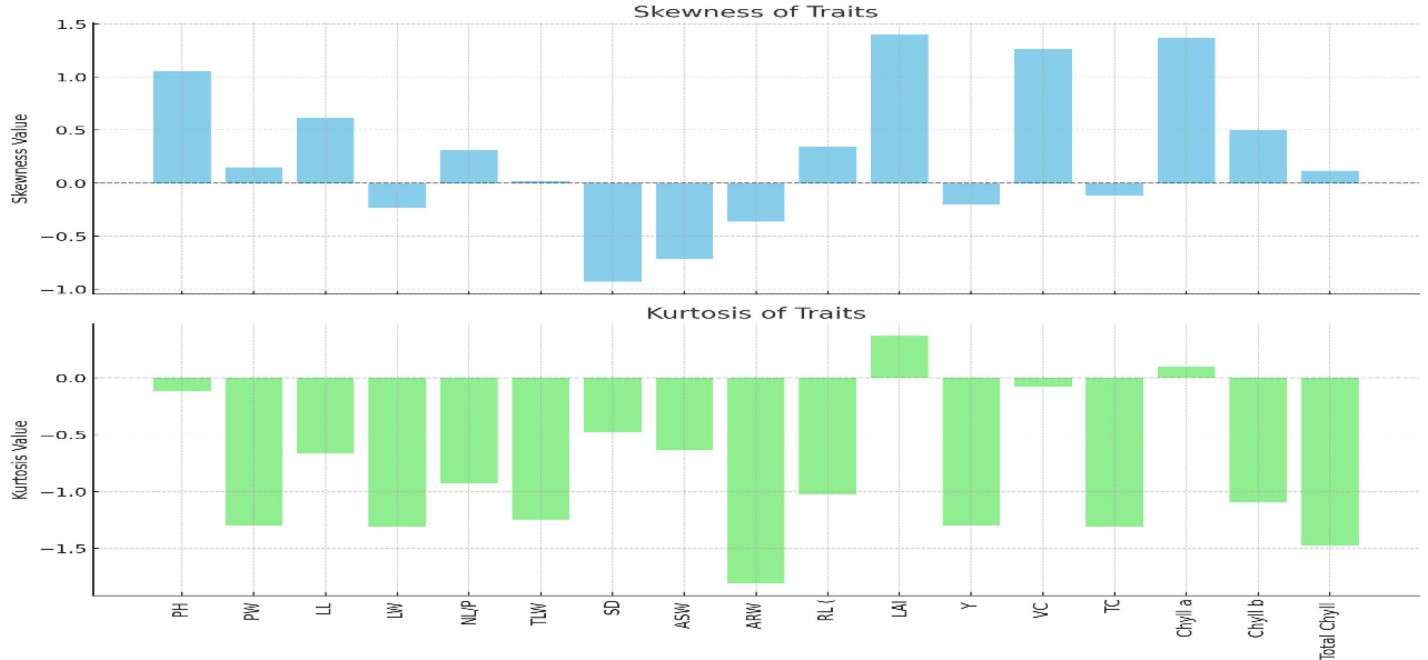

**Fig 7. Skewness and Kurtosis analysis of different traits of *Lactuca sativa*.**

## 4. Discussion

The positive skewness seen in aeroponics measures like plant weight (PW) and leaf length (LL) indicates that environmental conditions influence growth variability, meaning that a greater percentage of plants are smaller. According to recent research on aeroponic lettuce development, this might be the consequence of restricted nutrient distribution or insufficient light availability [40]. On the other hand, characteristics such as aerial root weight (ARW) and visual chlorophyll content (VC) show negative skewness, indicating that certain plants perform better, most likely as a result of increased nutrient uptake efficiency in the aeroponic setting [41].

The fact that most attributes have negative kurtosis suggests that lettuce plants respond consistently, reducing the number of extreme outliers. This points to the stability of growth in aeroponics, providing a controlled environment conducive to uniform development, which has been supported by findings in aeroponic studies focusing on growth consistency [Li et al., 2024].

For breeding programs seeking consistent root development in aeroponics, positive kurtosis in the instance of root length (RL) indicates that the trait is stable and centered around a mean. Aeroponics exhibits a platykurtic distribution in contrast to soil-based or hydroponic systems, promoting consistent growth for numerous attributes [Zhang et al., 2023]. Aeroponic system modifications, such as alterations to illumination or nutrient delivery schedules, can be guided by the skewness and kurtosis patterns. Large-scale aeroponic production can improve trait consistency and yield potential by concentrating on growth uniformity and aiming for traits with stable distributions (such as leaf area index (LAI) and RL).

Analyzing several lettuce types using PCA in the context of aeroponics provides valuable insights about growth patterns and performance. Compared to conventional hydroponics or soil-based techniques, aeroponics—a soilless growing technique in which plant roots are suspended in the air and misted with nutrients—has been demonstrated to accelerate development and increase nutrient absorption [42]. According to recent research, the initial main components of PCA may reflect the notable differences in biomass output, nutritional content, and overall growth efficiency across lettuce cultivars cultivated in aeroponic systems [43].

According to Wang et al. [41], lettuce types cultivated in aeroponic systems were more resilient to environmental stressors like nutrient changes than those cultivated in soil or hydroponic systems. The observed variance that the PCA analysis's first few principal components capture may be influenced by these variations. Furthermore, the cumulative variance explained by the first two principal components (86.13%) shows how some important factors, like environmental conditions, nutrient uptake, and the genetic variability of the lettuce varieties, significantly influence the results of growth in aeroponic systems [40].

The genetic variety of the lettuce varieties is shown by the clustering. Bingo and Summer Star are closely related, which may indicate that they have similar genetic compositions or environmental adaptations. This observation is consistent with results from a recent study that evaluated genetic variability among lettuce cultivars using high-throughput SNP arrays, forming unique groups according to genetic similarity [27].

The findings show that aeroponics can be a very effective way to grow lettuce, but that in order to maximize yield, it can be crucial to comprehend the precise performance variations among cultivars. Future research should examine the interactions between various kinds and aeroponic systems in more detail, possibly using multi-variable analysis to find other variables influencing growth patterns.

## 4.0 Conclusion

This study on lettuce cultivation in aeroponic system demonstrated significant genetic and environmental variability in growth and quality features among different varieties. Both genetic and environmental factors have a significant impact on important parameters as plant height, weight, total leaf weight, and chlorophyll content. The significance of characteristics in maximizing growth and yield is highlighted by strong relationships between them, such as plant height with chlorophyll content and leaf area index.

According to principal component analysis (PCA), characteristics such as plant height, leaf morphology, and chlorophyll content explain more than 86% of the variance, indicating that these characteristics are essential for enhancing aeroponic biomass production. The most promising traits for improvement were those like chlorophyll concentration, especially chlorophyll a, which presented chances for focused breeding. Higher yield and total leaf weight were shown by varieties including Summer Star and Bingo, which made them ideal for aeroponic systems. These results highlight how aeroponics might boost lettuce output and lay the groundwork for further studies on how to best optimize environmental factors and trait interactions for higher yield [44,45,46,47,48,49,50,51,4,52].

## Author contributions

**Conceptualization:** Anand Sahil, S. R. Singh.

**Data curation:** Anand Sahil, S. R. Singh.

**Investigation:** Anand Sahil.

**Methodology:** Anand Sahil, S. R. Singh.

**Visualization:** Lalu Prasad Yadav.

**Writing – original draft:** Anand Sahil, Lalu Prasad Yadav.

**Writing – review & editing:** S. R. Singh, Lalu Prasad Yadav, A. K. Singh, Vikas Yadav, Sanjay Kumar, N. K. Meena, Prashant Kaushik, Prakash Mahala.

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
