## [Decision Letter · Decision Letter 0]

5 Jan 2026

Dear Dr. Yadav,

Thank you for submitting your manuscript to PLOS ONE. After careful consideration, we feel that it has merit but does not fully meet PLOS ONE’s publication criteria as it currently stands. Therefore, we invite you to submit a revised version of the manuscript that addresses the points raised during the review process.

We look forward to receiving your revised manuscript.

Kind regards,

Eugenio Llorens

Academic Editor

PLOS One

Journal Requirements:

Reviewers' comments:

Reviewer's Responses to Questions

**Comments to the Author**

1. Is the manuscript technically sound, and do the data support the conclusions?

Reviewer #1: Yes

Reviewer #2: Yes

2. Has the statistical analysis been performed appropriately and rigorously?

Reviewer #1: Yes

Reviewer #2: Yes

3. Have the authors made all data underlying the findings in their manuscript fully available?

Reviewer #1: Yes

Reviewer #2: Yes

4. Is the manuscript presented in an intelligible fashion and written in standard English?

Reviewer #1: Yes

Reviewer #2: Yes

Reviewer #1: The author should provide fully information about cultivars used in this study and also complete background about this cultivars

Also, he must provide detailed information about this cultivars in regular field to compare between yield as this method is effective or not

Also, this investigation must provide us about costs of this method to proof is it effective or not and the increase in yield

Also, author should explain in details why he choose this lettuce cultivars used in this investigation

Reviewer #2: Add degree symbol in figure 1. Eleventh ....? complete the word. Cite original references for methodology. Check fig 4. SOme of the figures are not clearly visible. Please make them clear. Language should be polished. Introduction part can be reduced.

**Do you want your identity to be public for this peer review?** For information about this choice, including consent withdrawal, please see our Privacy Policy

Reviewer #1: **Yes:** Khaled M.H. Abd El Salam

Reviewer #2: No

---

## [Author Response · Author response to Decision Letter 1]

3 Feb 2026

Review Comments to the Author

We sincerely thank the editor and reviewers for overall insightful comments and constructive feedback to improve the MS.

Reviewer #1: The author should provide fully information about cultivars used in this study and also complete background about cultivars

Reply

We sincerely thank the reviewer for this valuable suggestion. We agree that providing clearer background information on the lettuce cultivars improves the scientific context and reproducibility of the study. In response, we have expanded the description of the cultivars used in this experiment.

Also, he must provide detailed information about this cultivars in regular field to compare between yield as this method is effective or not

Reply

We thank the reviewer for this important observation. We agree that comparing aeroponic performance with conventional field cultivation helps in better understanding the effectiveness of aeroponic systems.

However, the present study was specifically designed to evaluate varietal performance and trait interrelationships under controlled aeroponic conditions, and parallel open-field experiments were not conducted during the same season and location.

Also, this investigation must provide us about costs of this method to proof is it effective or not and the increase in yield

Reply

We sincerely thank the reviewer for this insightful comment. We agree that economic considerations are important when assessing the practical effectiveness of aeroponic systems.

The present study was designed as a preliminary investigation focusing on varietal performance, trait interrelationships, and biological productivity of lettuce under aeroponic conditions. Detailed techno-economic and cost–benefit analyses were therefore beyond the scope of this work and will be addressed in future research.

Also, author should explain in details why he choose this lettuce cultivars used in this investigation

Reply

We sincerely thank the reviewer for this constructive comment. We agree that a clear justification for cultivar selection strengthens the scientific rationale of the study.

In response, we have expanded the explanation for cultivar selection to clearly articulate why these specific lettuce cultivars were chosen for evaluation under aeroponic conditions.

Reviewer #2: Add degree symbol in figure 1. Eleventh ....? complete the word. Cite original references for methodology. Check fig 4. SOme of the figures are not clearly visible. Please make them clear. Language should be polished. Introduction part can be reduced.

Response

We thank the reviewer for pointing out this oversight.

Figure 1 has been corrected to include the degree (°) symbol.

The typographical error has been corrected at the indicated location in the manuscript.

Relevant original and standard references (e.g., AOAC, Fisher, Hotelling, Pearson, Lush, and published aeroponic studies) have been cited in the Materials and Methods section.

Figures have been revised with improved resolution

The Introduction has been shortened and refined to improve focus and conciseness.

---

## [Decision Letter · Decision Letter 1]

3 Mar 2026

Aeroponic Cultivation of Lettuce: Unravelling Varietal Performance and Trait Interrelationships for Enhanced Productivity

PONE-D-25-63594R1

Dear Dr. Yadav,

We’re pleased to inform you that your manuscript has been judged scientifically suitable for publication and will be formally accepted for publication once it meets all outstanding technical requirements.

Kind regards,

Eugenio Llorens

Academic Editor

PLOS One

Additional Editor Comments (optional):

Reviewers' comments:

Reviewer's Responses to Questions

**Comments to the Author**

Reviewer #2: All comments have been addressed

2. Is the manuscript technically sound, and do the data support the conclusions?

Reviewer #2: Yes

3. Has the statistical analysis been performed appropriately and rigorously?

Reviewer #2: Yes

4. Have the authors made all data underlying the findings in their manuscript fully available?

Reviewer #2: Yes

5. Is the manuscript presented in an intelligible fashion and written in standard English?

Reviewer #2: Yes

Reviewer #2: (No Response)

**Do you want your identity to be public for this peer review?** For information about this choice, including consent withdrawal, please see our Privacy Policy

Reviewer #2: No

---

## [Editor Report · Acceptance letter]

PONE-D-25-63594R1

PLOS One

Dear Dr. Yadav,

I'm pleased to inform you that your manuscript has been deemed suitable for publication in PLOS One. Congratulations! Your manuscript is now being handed over to our production team.

Kind regards,

on behalf of

Dr. Eugenio Llorens

Academic Editor

PLOS One